# Early Weaning Inhibits Intestinal Stem Cell Expansion to Disrupt the Intestinal Integrity of Duroc Piglets via Regulating the Keap1/Nrf2 Signaling

**DOI:** 10.3390/antiox13101188

**Published:** 2024-09-30

**Authors:** Ying-Chao Qin, Cheng-Long Jin, Ting-Cai Hu, Jia-Yi Zhou, Xiao-Fan Wang, Xiu-Qi Wang, Xiang-Feng Kong, Hui-Chao Yan

**Affiliations:** 1State Key Laboratory of Swine and Poultry Breeding Industry, Guangdong Laboratory for Lingnan Modern Agriculture, Guangdong Provincial Key Laboratory of Animal Nutrition Control, National Engineering Research Center for Breeding Swine Industry, College of Animal Science, South China Agricultural University, Guangzhou 510642, China; yingchaoqin1994@163.com (Y.-C.Q.); htc510@stu.scau.edu.cn (T.-C.H.); 31100020@scau.edu.cn (J.-Y.Z.); xxw033@scau.edu.cn (X.-F.W.); xqwang@scau.edu.cn (X.-Q.W.); 2Key Laboratory of Animal Nutrition and Feed Science in South China, State Key Laboratory of Swine and Poultry Breeding Industry, Guangdong Provincial Key Laboratory of Animal Breeding and Nutrition, Institute of Animal Science, Guangdong Academy of Agricultural Sciences, Ministry of Agriculture and Rural Affairs, Guangzhou 510640, China; jinchenglong1992@163.com; 3Key Laboratory of Agro-Ecological Processes in Subtropical Region, Hunan Provincial Key Laboratory of Animal Nutritional Physiology and Metabolic Process, National Engineering Laboratory for Pollution Control and Waste Utilization in Livestock and Poultry Production, Institute of Subtropical Agriculture, Chinese Academy of Sciences, Changsha 410125, China

**Keywords:** early weaning stress, intestinal stem cells, Duroc piglets, Taoyuan Black piglets, Keap1/Nrf2 pathway

## Abstract

There are different stress resistance among different breeds of pigs. Changes in intestinal stem cells (ISCs) are still unclear among various breeds of piglets after early weaning. In the current study, Taoyuan Black and Duroc piglets were slaughtered at 21 days of age (early weaning day) and 24 days of age (3 days after early weaning) for 10 piglets in each group. The results showed that the rate of ISC-driven epithelial renewal in local Taoyuan Black pigs hardly changed after weaning for 3 days. However, weaning stress significantly reduced the weight of the duodenum and jejunum in Duroc piglets. Meanwhile, the jejunal villus height, tight junction-related proteins (ZO-1, Occludin, and Claudin1), as well as the trans-epithelial electrical resistance (TEER) values, were down-regulated after weaning for 3 days in Duroc piglets. Moreover, compared with Unweaned Duroc piglets, the numbers of Olfm4^+^ ISC cells, PCNA^+^ mitotic cells, SOX9^+^ secretory progenitor cells, and Villin^+^ absorptive cells in the jejunum were reduced significantly 3 days after weaning. And ex vivo jejunal crypt-derived organoids exhibited growth disadvantages in weaned Duroc piglets. Notably, the Keap1/Nrf2 signaling activities and the expression of HO-1 were significantly depressed in weaned Duroc piglets compared to Unweaned Duroc piglets. Thus, we can conclude that ISCs of Duroc piglets were more sensitive to weaning stress injury than Taoyuan Black piglets, and Keap1/Nrf2 signaling is involved in this process.

## 1. Introduction

Intestinal stem cells (ISCs) drive self-renewal and damage repair of the intestinal epithelium [1]. During homeostasis, Lgr5^+^ intestinal stem cells constantly generate enterocytes (Paneth, enteroendocrine, goblet, and absorptive cells) to maintain the self-renewal of the crypt-villus axis and repair the intestinal epithelium after damage [2,3].

The weaning time for piglets under natural conditions is 17 weeks of age. However, in modern intensive pig production, the weaning time has advanced to 3–4 weeks of age to improve the efficiency of pig breeding [4,5]. The intestinal function of piglets is incomplete at the time of early weaning. Therefore, early weaning can easily change intestinal morphology and physiological function, decreasing digestion and absorption [6]. As a rapidly renewing tissue, the intestinal epithelium can regenerate rapidly after damage via tightly controlled mechanisms, which play a central role in nutrient absorption and avoiding the invasion of external adverse factors [7,8]. Accordingly, the rapid renewal of the intestine in weaned piglets is important to resist weaning stress and to restore growth performance after weaning.

China is rich in indigenous pig genetic resources, and the indigenous landraces have formed distinct clades [9]. Different breeds of pigs have different phenotypes and exhibit different resistance to viruses, bacteria, and various adverse factors [10,11,12]. In particular, the stress resistance of Chinese indigenous is significantly different from that of western pig breeds [13]. Taoyuan Black pig, the representative of the native breed of Hunan province in China, has a higher tolerance to dietary fiber than the Duroc pigs [14]. Increasing evidence implies that weaning can trigger weaning stress and redox imbalance in piglets [15,16]. However, the ability of Taoyuan Black pigs to resist intestinal damage caused by weaning stress remains unclear. Therefore, decoding the coupling of weaning stress to ISC expansion and intestinal integrity will be valuable for avoiding weaning stress-induced damage to intestinal epithelial function.

Several studies have suggested that dietary supplementation with selenium-enriched yeast or stevia residue extract can increase the antioxidant capacity and reduce piglet diarrhea induced by early weaning [17,18]. A generally accepted view is that an oxidative stress-induced transcription factor Nrf2 plays a major role in maintaining adaptive homeostasis and adaptive responses to oxidative stress, and Keap1 can regulate Nrf2 [19,20]. Nevertheless, it remains unclear whether different piglets have different resistance to weaning stress and whether oxidative stress is involved in early weaning-induced intestinal injury. Using piglets and intestinal organoids (IOs) as models, we identified that Taoyuan Black avoids oxidative stress induced by early weaning via maintaining Keap1/Nrf2 signaling activity, thereby maintaining ISC expansion activity after early weaning and maintaining intestinal integrity.

## 2. Materials and Methods

### 2.1. Piglets Treatment Regimen

The local ethical review committee (Animal Ethics Committee of South China Agricultural University) sanctioned the experiments. A total of 40 Taoyuan Black and Duroc piglets (20 piglets from each breed and half male and half female; the male piglets were castrated at 5 days of age) with similar weights were selected on the weaning day (21 days of age). Taoyuan Black and Duroc piglets were slaughtered with 10 pigs each breed on weaning day and 3 days after weaning. Thus, all piglets were divided into 4 groups: Taoyuan Black Unweaned (slaughtered on the weaning day), Taoyuan Black weaned (slaughtered on 3 days after weaning), Duroc Unweaned (slaughtered on the weaning day), and Duroc weaned (slaughtered on 3 days after weaning). We separated each small intestine segment with scissors and washed them with PBS. Then, the length and weight of the small intestine were measured. The intestinal mucosa was further stripped from longitudinal muscle with forceps and weighed.

### 2.2. Hematoxylin–Eosin (H&E) and Immunohistochemistry (IHC) Staining

The jejunum fixed with 4% paraformaldehyde was dehydrated with graded ethanol (70%, 80%, 90%, and 100%) and embedded with paraffin after permeated with xylene. Paraffin-embedded tissue blocks were sections into 3–5 μm sections with the Leica microtom. Paraffin sections were stained with hematoxylin and then eosin and monitored under a light microscope at 200× magnification (Leica, Wetzlar, Germany).

The paraffin sections for IHC staining of Occludin, Olfm4, SOX9, PCNA, KRT20, Villin, C-Caspase3, SOD1, p-Nrf2 and HO-1 were performed, respectively. Antibody information is shown in Appendix A;Representative sections were photographed using an inverted fluorescence microscope (Ti2-E, Nikon, Tokyo, Japan). Fluorescence signals were quantified by ImageJ (version 1.8.0 112, National Institutes of Health, Bethesda, MD, USA), and each respective Unweaned group was set at 1 (Taoyuan Black weaned vs. Unweaned and Duroc weaned vs. Unweaned).

### 2.3. Western Blotting (WB)

Jejunum tissues were lysed by RIPA Lysis Buffer (#DB258, MIKX), and equal amounts of sample lysates and color pre-stained protein marker (M222, GenStar) were separated on SDS/PAGE gels and transferred on PVDF membranes (Millipore, Burlington, VT, USA). The proteins were visualized using the Femto Sensitive Plus ECL chemiluminescence kit (MK-S500, MIKX). Then, the band densities were quantified with ImageJ. Each respective Unweaned group was set at 1 (Taoyuan Black weaned vs. Unweaned and Duroc weaned vs. Unweaned).

### 2.4. Transepithelial Electrical Resistance (TEER) Value

The agar powder is heated and dissolved in the KCl solution to form a transparent gelatinous liquid, and the agar gel solution is poured into the electrode head from the tip of the electrode head to prepare a salt bridge. After connecting the salt bridge to the electrode head, install the Ussing Chamber. Once the instrument is stable, zero the instrument.

We spread the jejunal tissue on the Ussing chamber and read the resistance values of the intestinal tissue. After the data are stable, the resistance value is collected, and the resistance value is converted into the resistance value of intestinal tissue per unit area according to the size of the upper sample area, which is the TEER.

### 2.5. Determination of Glutathione Peroxidase (GSH-Px) Activity

The activities of GSH-Px (#A005-1-2, Jiancheng Bioengineering Institute) in jejunum and serum were measured using commercial kits.

### 2.6. Determination of Malondialdehyde (MDA) Levels in Serum and Jejunum

MDA levels of serum and jejunum were assessed based on thiobarbituric acid test by using commercial kits (#A003-1-2, JianchengBioengineering Institute).

### 2.7. Culture of Intestinal Organoids (IOs) and Immunofluorescence Staining

The jejuna crypt of the piglets in each group was isolated by EDTA chelation and cultured in three-dimensional (3D) stereoscopic. Approximately 40 crypts are inoculated in 25 μL Matrigel (BD Biosciences, San Jose, CA, USA).

We divided 12 16-day-old male C57 mice into two groups: 4 in the control group and 8 in the early weaning group. Isolated the jejunal crypts of the control group and the early weaning group for ex vivo culture after three days of weaning to construct an early weaning model in organoids. We treated the IOs in the early weaning group with 10 μmol/L Sulforaphane (Nrf2 activator) for four days and observed the growth trend of organoids in each group.

The forming efficiency of organoid was calculated as the percentage of the colony number over the number of crypts seeded the next day. The organoid budding efficiency was calculated as the percentage of budding organoids over the colony number. Organoids were classified according to the number of organoid branches. The budding number was calculated according to the ratios of various types of organoids in the colony and the branching coefficient differences in the proportions of organoids among all groups. The surface area was measured with NIS-Elements Viewer 5.21 software (Nikon, Tokyo, Japan).

Crypt-derived organoids were fixed with paraformaldehyde for 30 min at 4 °C, permeabilized with a 0.5% Triton X-100 solution at 4 °C for 10 min, and blocked in fetal bovine serum for 60 min. Then, the cells were incubated with primary antibodies (1:200 dilution) for 8 h at 4 °C and with a secondary antibody (1:500 dilution) for 2 h at room temperature. The nuclei were stained with DAPI (1:1000 dilution) for 10 min at room temperature. Images were obtained using a confocal microscope (AX, Nikon, Tokyo, Japan).

### 2.8. Automated Capillary Western Blotting (WES)

IOs were lysed in RIPA buffer for 30 min at 4 °C. The protein concentration of the supernatant was determined with the BCA Protein Assay Reagent kit after centrifugation at 12,000× *g*. According to the user guide, the protein expression of intestinal organoid samples was determined with a WES instrument. The instrument ran automatically after the lysates, antibodies, and buffer were added to the assay plate, and the data were analyzed with Compass software 4.0 (ProteinSimple, San Jose, CA, USA). Each respective Un-weaned group was set at 1.

### 2.9. Statistical Analysis

The data are presented as the mean ± SEM. All statistical analyses were performed with GraphPad Prism 8.0 software and IBM SPSS Statistics version 20. Independent *t*-tests were used for statistical analysis. Significance was set at *p* < 0.05 for all experiments.

## 3. Results

### 3.1. Early Weaning Disturbs the Crypt-Villus Axis of Duroc Piglets

To evaluate the ability of different piglets to resist intestinal damage induced by early weaning stress, we used Taoyuan Black and Duroc piglets to explore the sensitivity to early weaning stress in different breeds. Taoyuan Black piglets and Duroc piglets were weaned and slaughtered on the weaning day and 3 days after weaning with 10 piglets each (Taoyuan Unweaned; Taoyuan Weaned; Duroc Unweaned and Duroc Weaned).

Early weaning in Duroc piglets reduced the weights of duodenum and jejunum, as well as mucosal weight, and tended to decrease the intestinal and mucosal weights in the ileum (Figure 1A,B). However, the small intestine of Taoyuan Black piglets did not change after early weaning. The jejunum represents a major part of the small intestine, the most active dietary absorption region [21,22]. After weaning, the H&E Staining paraffin sections of the jejunum were analyzed to determine the crypt–villus axis structure. Early weaning exhibited a more significant decrease in villus height (Figure 1C,D) and reduced the ratio of the villus to crypt (Figure 1F). Thus, early weaning can perturb intestinal development in Duroc and Taoyuan Black piglets can resist the intestinal damage caused by early weaning.

### 3.2. Early Weaning Disrupts the Jejunal Integrity of Duroc Piglets

The alteration in intestinal permeability and changes in tight junction proteins are the signs of a perturbed barrier function. Thus, piglets were sacrificed, and their jejunum intestines were collected before measuring the intestinal permeability using an Ussing chamber. As shown in Figure 2A, the jejunal TEER value of Duroc piglets was significantly decreased after weaning 3 days, indicating that jejunal permeability was upregulated. However, early weaning did not inhibit jejunal TEER value in Taoyuan Black pigs (*p* = 0.09).

We also examined the jejunum’s expression levels of tight junction proteins (ZO-1, Occludin, and Claudin 1) to validate the result of the Ussing chamber. The level of tight junction protein was significantly down-regulated in Duroc piglets after weaning (Figure 2B–E). The above results suggest that the Taoyuan Black piglets resisted the intestinal tight junction damage induced by early weaning. In contrast, Duroc piglets were more sensitive to weaning stress.

We monitored ISC division and differentiation into functional cells along the crypt-villus axis to explore the influence of intestinal self-renewal on early weaning stress. Early weaning decreased the numbers of ISCs (Olfm4^+^ cells), secretory progenitor cells (SOX9^+^ cells), and proliferating cells (PCNA^+^ cells) in the crypts, accelerated the occurrence of cell apoptosis (marked by C-Caspase3) and retarded ISC differentiation (KRT20^+^ cells) into enterocytes (Villin^+^ cells) (Figure 3). Early weaning did not disrupt the self-renewal of the Taoyuan Black piglet intestines. Thus, early weaning blocked the orderly renewal of jejunum functional cells in Duroc piglets, but Taoyuan Black piglets can resist the destruction of early weaning on intestinal development.

### 3.3. The Maintenance of Keap1/Nrf2 Signal Activity Can Prevent the Oxidative Stress Induced by Early Weaning in Taoyuan Black Piglets

To investigate the effects of weaning stress on the oxidation-antioxidant status of piglets, we detect related indicators of oxidative stress. IHC of SOD1 shows that the SOD1 level in the jejunum of Duroc piglets decreased significantly after early weaning (Figure 4A,B). Likewise, the GSH-Px activity was reduced in the jejunum and the serum of Duroc piglets (Figure 4C,D). MDA contents in jejunum and serum were increased (Figure 4E,F). Intestinal oxidative stress in Duroc piglets occurred at 3 days after 3 weaning. In contrast to Duroc piglets, Taoyuan Black pigs have the potential to resist oxidative stress induced by weaning stress.

To investigate the relationship between weaning stress and whether it is related to Keap1/Nrf2 signaling, we detected the protein (p-Nrf2 and HO-1) by IHC and WB. Early weaning significantly downregulated the levels of p-Nrf2 and HO-1 in the Duroc piglets. However, Taoyuan Black pigs did not change before and after weaning (Figure 4G–K).

### 3.4. Early Weaning Decreases Growth Advantages in IOs and Weakens Keap1/Nrf2 Signaling in Duroc Piglets

To determine the effect of early weaning on intestinal stem cell expansion, we tested the jejunal organoids of Taoyuan Black and Duroc piglets before and after early weaning. The results indicated that early weaning attenuated the growth advantages of IOS (Figure 5A). Specifically, we observed a significant decrease in forming efficiency (Figure 5B), surface area (Figure 5C), and budding efficiency (Figure 5D) of IOs in Duroc piglets after weaning 3 days. In addition, the number of organoids with a budding number greater than 5 decreased significantly in Duroc piglets after weaning 3 days (Figure 5E,F). In addition, the results of WES combined with organoid immunofluorescence results showed that the p-Nrf2 levels were inhibited along with early weaning in IOs of Duroc piglets (Figure 5G–I). These data together suggest that early weaning inhibits Keap1/Nrf2 signaling-driven ISC regenerative renewal in Duroc piglets (Figure 6).

## 4. Discussion

China has diverse indigenous pigs and abundant resources [9]. Compared to European pig breeds, Chinese native breeds are renowned for their good adaptability to the local environment [23], high meat quality [24], and high forage tolerance [25]. Taoyuan Black, an indigenous pig breed from Hunan province in China, is more tolerant to dietary fiber and has stronger immune function than the Duroc pigs [14,26]. We speculate that Taoyuan black pigs will likely be highly resilient to weaning stress. However, there are no other studies on the stress resistance of Taoyuan Black pigs.

The early weaning strategy has been widely used in pig production to improve sows’ reproductive performance and reduce feed costs per pig to increase the efficiency of raising pigs [16]. The integrity of the intestinal mucosa structure is the basis for normal digestion and absorption of nutrients that support normal body growth. However, early weaning at around 21 days of age often leads to intestinal architecture impairment and malabsorption, finally resulting in intestinal disorders and retarded growth, even death from diarrheal in most intensive farming of commercial piglets, as gastrointestinal digestive systems not yet mature at early weaning [27,28]. In this study, we found that the united weight of tissue and mucosa in jejunum and ileum decreased significantly at 3 days after weaning, and the weight of ileum and mucosa membrane per unit length had a decreasing trend in Duroc but not Taoyuan Black piglets compared with weaning day. We weaned mice at 16 days of age to construct an early weaning model. The results show that the body weight of early-weaned mice reached the lowest levels at 3 days after weaning (Appendix A). Since the main site for nutrient absorption in mammals is the jejunum [29], we further analyzed jejunal morphology by H&E staining. The crypt-villus axis, especially the villi structure, was damaged 3 days after early weaning in Duroc piglets. Like Duroc piglets, the jejunum of the mice was also severely damaged after early weaning (Appendix A). Thus, we preliminarily determined that Taoyuan Black pigs were resistant to the intestinal damage caused by weaning.

The intestinal epithelium is a single-cell layer consisting of different types of cells with multiple functions, responsible for absorbing nutrients and forming a barrier between the external environment and the body [30]. Modifying TEER values indicates disturbances in the transcellular permeability of ions and the intestinal epithelium barrier [31]. As part of the intestinal barrier, tight junction proteins seal the space between the different intestinal epithelial cells to block intestinal damage by adverse factors such as potential pathogens and toxins [32]. Our study confirmed that early weaning significantly increased jejunal permeability and decreased the levels of tight junction proteins in Duroc piglets. The jejunal permeability of mice also increased significantly at 3 days after early weaning (Appendix A). To detect the expression of each marker protein, we found that Taoyuan Black pigs were resistant to weaning-induced changes in ISC proportion, mitotic cell number, and functional cell composition, which was different from Duroc piglets and mice (Appendix A). Thus, the resistance of Taoyuan pigs is beneficial for maintaining intestinal structure and function after weaning.

It is well known that oxidative stress is directly involved in many injuries. For example, the intestinal toxins deoxynivalenol [33], bacterial infection (Enterotoxigenic *Escherichia coli*) [34], and porcine epidemic diarrhea [35] can induce oxidative stress and injury in pigs. Generally speaking, piglets are subjected to nutritional, environmental, and psychological stress during the weaning period, leading to post-weaning diarrhea syndrome [36]. Therefore, we can speculate that weaning syndrome, as comprehensive traumatic stress, is likely to be related to oxidative stress. Thus, we detected the oxidative stress indexes in the jejunum (SOD1, GSH-Px, and MDA) and serum (GSH-Px, and MDA) before and after weaning of Taoyuan Black and Duroc piglets and found that early weaning induced an imbalance of oxidative stress indexes in the jejunum and serum in Duroc but not in Taoyuan Black piglets.

The intestinal epithelium is a prototypical rapidly renewing tissue driven by ISCs to maintain intestinal epithelial structure and function [37]. Literature has shown that intestinal damage caused by harmful factors is related to the loss of ISC expansion activity [38], and the nutrients or bioactive substances can promote intestinal development by increasing ISC activity and reversing intestinal damage caused by adverse factors [39]. Therefore, we can speculate that maintaining the orderly regeneration and self-renewal ability of ISCs in piglets is conducive to resisting intestinal damage induced by weaning stress. The results of jejunal stem cell culture before and after weaning demonstrated that early weaning interfered with the growth advantage of ISCs by reducing the formation efficiency and area of organoids and inhibiting the development of organoid buds in Duroc piglets. Similar to the Duroc piglets, we also found that early weaning inhibited the proliferation of ISC in the jejunum ex vivo in mice (Appendix A). Taken together, early weaning inhibits ISC expansion in Duroc piglets and mice. However, Taoyuan Black piglets can resist ISC expansion impaired by early weaning.

Oxidative stress can lead to the oxidation of biomolecules (nucleic acids, lipids, and proteins) or the activation of inflammatory signaling pathways, activating several transcription factors, or the dysregulation of gene and protein expression followed by inflammatory response and immune dysfunction [40,41,42]. The mild inflammatory response is beneficial in resisting the stimulation and damage of various pathogenic factors. Excessive inflammatory factors will make the inflammatory response lose its original coordination and aggravate the damage induced by adverse factors [43]. Different breeds of pigs have different tolerances to weaning stress, possibly due to the differences in inflammatory factor levels or sensitivity to inflammatory factors before and after weaning.

Nrf2 is a key transcription factor in the intracellular antioxidant system and maintains redox homeostasis by activating downstream enzymes [44]. At the same time, Nrf2 can also suppress macrophage inflammatory response by blocking proinflammatory cytokine transcription [45]. Nrf2 deficiency in mice is more likely to lead to redox imbalance and is associated with the length of life [46]. Nrf2 interacts with Keap1, which suppresses Nrf2 activity and directs Nrf2 for proteasome degradation [47]. The phosphorylation of Nrf2 facilitates Keap1/Nrf2 dissociation and Nrf2 nuclear translocation and initiates transcription of downstream enzymes such as HO-1 and SOD1 [48]. As an antioxidant enzyme, HO-1 reduces oxidative stress by metabolizing heme to biliverdin, Fe^2+^, and CO. These substances have been shown to have antioxidative and anti-inflammatory properties [49]. SOD consists of three enzymatic isoforms: cytosolic CuZn-SOD (SOD1), mitochondrial Mn-SOD (SOD2), and extracellular CuZn-SOD (SOD3) [50]. As the major SOD, SOD1 comprises 90% of total SOD and is involved in the removal of superoxide [51]. The present findings demonstrated that early weaning upregulated the Keap1 level and downregulated the expression of p-Nrf2 and the transcription of HO-1 and SOD1 in the jejunum and organoids of Duroc piglets. It is worth noting that none of the above changes appeared in the jejunum and organoids of Taoyuan Black piglets after weaning. Using mice as a research model, we found that up-regulation of Nrf2 signaling activity alleviated the inhibitory effect of early weaning on the expansion of mouse jejunal stem cells by processing 10 μmol/L Sulforaphane (Nrf2 activator) (Appendix A). These results suggest that Nrf2 mediates weaning stress-induced intestinal injury, which provided new insights into the analysis of stress resistance of indigenous pigs in China.

## 5. Conclusions

The present study demonstrates that early weaning induced intestinal destruction and redox dyshomeostasis. Mechanistically, the weaning syndrome suppresses the Keap1/Nrf2 signaling pathway, reducing the production of antioxidants to accumulate ROS and destroy ISC regeneration, thereby leading to intestinal epithelial injury in Duroc piglets. It must be emphasized that the Taoyuan Black pigs exhibited a stronger tolerance to early weaning stress by maintaining the activity of Keap1/Nrf2 signaling (Figure 6). Our study provided the theoretical foundations for resistance to weaning stress through targeted breeding using local pig breeding resources.

## Figures and Tables

**Figure 1 antioxidants-13-01188-f001:**
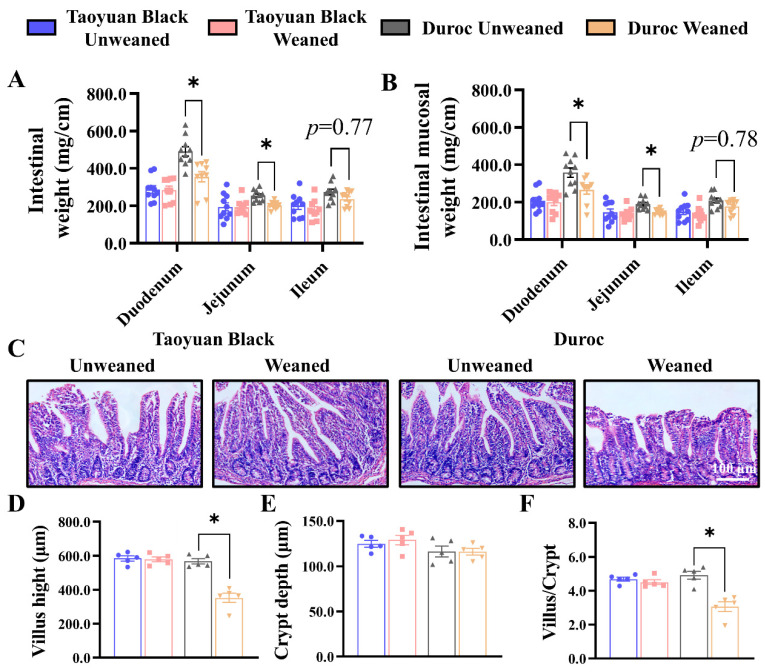
Early weaning damages the intestinal structure of Duroc piglets. (**A**,**B**) Intestinal weight per centimeter (**A**) and intestinal mucosal weight per centimeter (**B**) were measured (*n* = 10 piglets). (**C**) Representative images of H&E staining in the jejunum of piglets are shown. (**D**–**F**) The results of the statistical analysis of villus height (**D**), crypt depth (**E**), and the ratio of the villus to crypt (**F**), *n* = 4 piglets. The results are expressed as the mean ± SEM. * *p* < 0.05.

**Figure 2 antioxidants-13-01188-f002:**
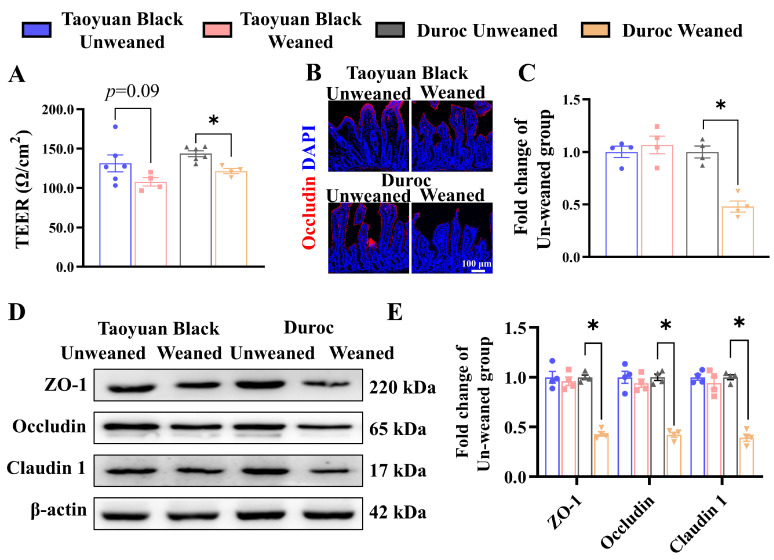
Early weaning destroys jejunal barrier integrity in Duroc piglets. (**A**) The TEER of jejunum. (**B**,**C**) Occludin protein expression levels were detected by immunohistochemical staining. (**D**,**E**) Tight junction protein expression levels were detected by Western blotting. The results are expressed as the mean ± SEM (*n* = 4 piglets). Each respective Unweaned group was set at 1. * *p* < 0.05.

**Figure 3 antioxidants-13-01188-f003:**
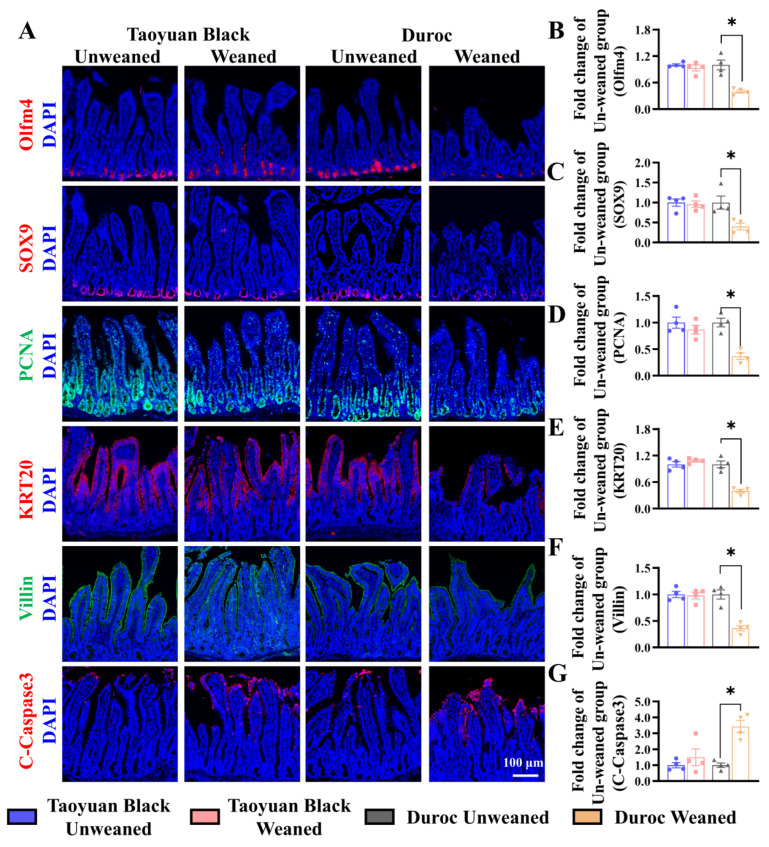
Early weaning disrupts intestinal growth in the jejuna of Duroc piglets. (**A**–**G**) Representative images and the results of statistical analysis of immunohistochemical staining with Olfm4, SOX9, PCNA, KRT20, Villin, and Cleaved Caspase3 (C-Caspase3) in the jejuna of piglets. The results are expressed as the mean ± SEM (*n* = 4 piglets). Each respective Unweaned group was set at 1. * *p* < 0.05.

**Figure 4 antioxidants-13-01188-f004:**
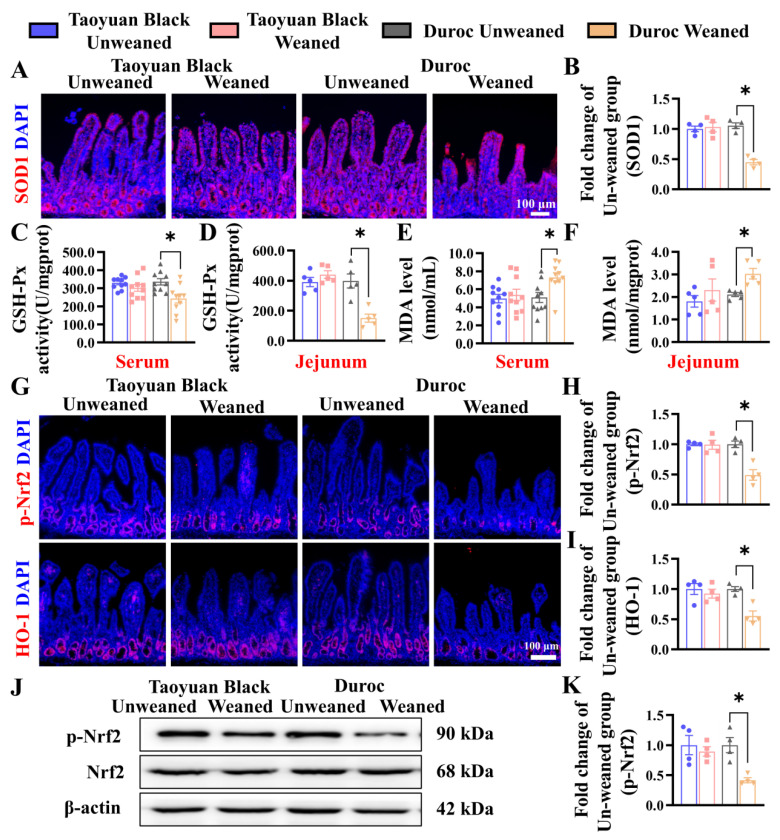
Early weaning induces oxidative stress and inhibits Keap1/Nrf2 signaling in the jejunum of Duroc piglets. (**A**,**B**) SOD1 protein expression levels were detected by immunohistochemical staining. (**C**–**F**) GSH-Px activity and MDA levels in serum and jejunum were detected (*n* = 10 piglets). (**G**–**I**) Keap1/Nrf2 signaling pathway-related proteins were analyzed in jejunum by immunohistochemical staining. (**J**,**K**) Keap1/Nrf2 signaling pathway-related proteins were analyzed in the jejunum by Western blotting. The results are expressed as the mean ± SEM (*n* = 4 piglets). Each respective Unweaned group was set at 1. * *p* < 0.05.

**Figure 5 antioxidants-13-01188-f005:**
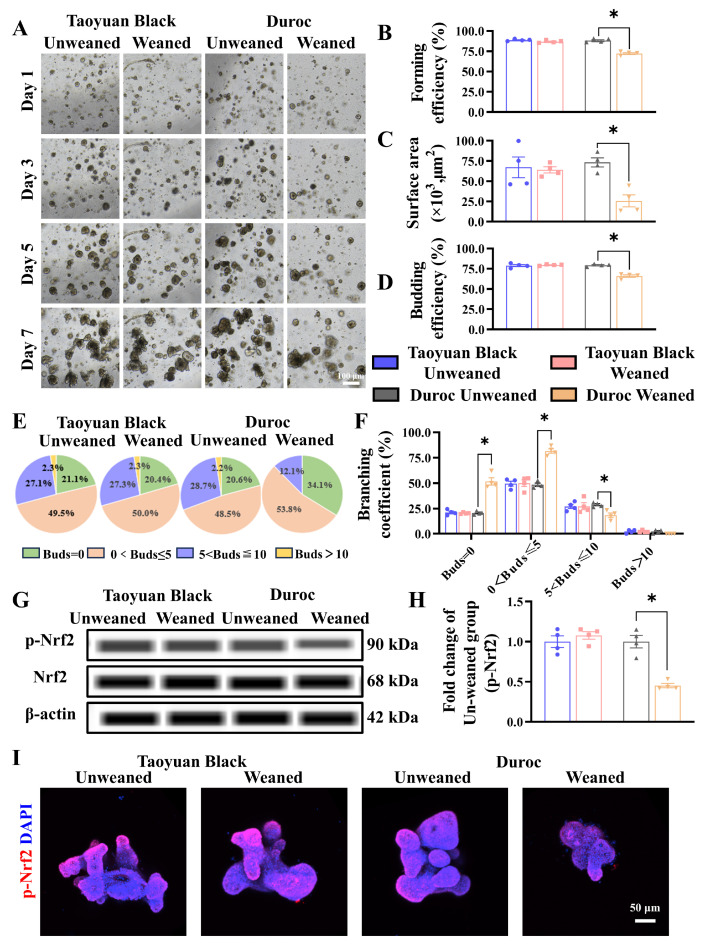
Early weaning inhibits intestinal stem cell proliferation and Keap1/Nrf2 signaling activity in the jejunum organoids of Duroc piglets ex vivo. (**A**) Representative images of intestinal organoids cultured from crypt cells in each group of piglets on days 1, 3, 5, and 7 are shown. (**B**–**F**) The organoid forming efficiency was measured in D3 (**B**); surface area (**C**), budding efficiency (**D**), bud number (**E**), and branching coefficient of organoids were measured in D7 (**F**). (**G**–**I**) Keap1/Nrf2 signaling pathway-related proteins were analyzed in the jejunum organoids by WES and Immunofluorescence combined with confocal. The results are expressed as the mean ± SEM (*n* = 5 piglets). Each respective Unweaned group was set at 1. * *p* < 0.05.

**Figure 6 antioxidants-13-01188-f006:**
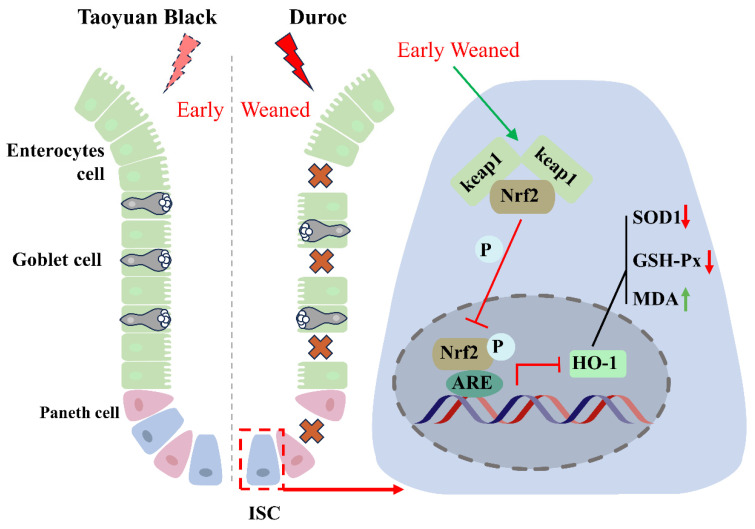
Early weaning inhibits intestinal stem cell proliferation via inhibiting Keap1/Nrf2 signaling activity in Duroc piglets. Crosses indicate that the intestinal epithelium is disrupted by early weaning. The red arrow indicates a decrease in the indicator, and the green arrow indicates an increase in the indicator. The T-symbol indicates inhibition of the associated signaling activity.

## Data Availability

The data that support the findings of this study are available upon reasonable request.

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
