# Peer review of "Early Weaning Inhibits Intestinal Stem Cell Expansion to Disrupt the Intestinal Integrity of Duroc Piglets via Regulating the Keap1/Nrf2 Signaling"

_antioxidants, 2024, doi:10.3390/antiox13101188_

Round 1

Reviewer 1 Report

Qin and colleagues conducted a comparative study of intestinal stem cells among breeds of Taoyuan Black and Duroc pigs with an emphasis on oxidative signaling via the Keap1/Nrf2 signaling pathway. A combination of experimental approaches included histology, western blot, transepithelial electrical resistance, and intestinal organoid growth. As a result of their findings, the authors conclude that, relative to the Taoyuan Black piglets, the intestinal stem cells of the Duroc piglets are more sensitive to oxidative stress during weaning.   

My detailed comments concern the presentation of Results as described above. Provided the experimental design for comparisons among two different piglet breeds, the appropriate analysis would actually entail a two-way ANOVA. Use of male and female animals needs to be specified as well.

Author Response

List of reviewer comments and revisions of the manuscript

Dear Reviewers,

First of all, many thanks for your invaluable comments on our manuscript entitled “Early weaning inhibits intestinal stem cell expansion to disrupt the intestinal integrity of Duroc piglets via regulating the Keap1/Nrf2 signaling”. We have revised the paper according to your  comments, as described below.

Thank you very much for your excellent comments on our manuscript. We were thrilled to revise our paper based on your comments. The changes are highlighted in RED in the revised manuscript.

Major comments:

Qin and colleagues conducted a comparative study of intestinal stem cells among breeds of Taoyuan Black and Duroc pigs with an emphasis on oxidative signaling via the Keap1/Nrf2 signaling pathway. A combination of experimental approaches included histology, western blot, transepithelial electrical resistance, and intestinal organoid growth. As a result of their findings, the authors conclude that, relative to the Taoyuan Black piglets, the intestinal stem cells of the Duroc piglets are more sensitive to oxidative stress during weaning.

Detail comments:

My detailed comments concern the presentation of Results as described above. Provided the experimental design for comparisons among two different piglet breeds, the appropriate analysis would actually entail a two-way ANOVA. Use of male and female animals needs to be specified as well.

Answer:

Thank you for your comments. First, we analyze the weight of small intestine and mucous per length of different breeds of pigs by two-way ANOVA, and the results are as follows in Table 1.

Table 1 Intestinal development differences of weaned piglets among different breeds before and after weaning

Table 1 Intestinal development differences of weaned piglets among different breeds before and after weaning

As shown in Table 1, Taoyuan Black pigs have a strong tolerance to weaning stress, and their intestinal weight did not change significantly before and after weaning, so they concealed the destructive effect of weaning stress on the jejunum. Thus, we used an independent T-test to count the intestinal weight of different breeds of pigs before and after weaning, as shown in Figure 1A-1B of the manuscript. There are no significant changes in the Jejunal weight of Taoyuan black pigs before and after weaning. In contrast, the jejunum and ileal weight per unit length of Duroc pigs decreased significantly after weaning, showing the different tolerance of different breeds of pigs to weaning stress. We also used an independent T-test to count the data in the following studies.

In addition, we describe the sex of the pigs used in our research in Materials and Methods 2.1. The ovaries did not develop during weaning, and the male piglets were castrated at 5 days of age. Therefore, we did not consider that gender differences had any effect on the results of this experiment. The sex ratio of piglets we selected is 1:1, which more realistically reflects the changes of piglets before and after weaning in actual production.

There are concerns encompassing analysis and illustration of results.

 (1) Please specify the number of males and females used across experiments and whether there are differences to note among biological sexes.

Answer:

We describe the sex of the pigs used in our research in Materials and Methods 2.1. The ovaries did not develop during weaning, and the male piglets were castrated at 5 days of age. Therefore, we did not consider that gender differences had any effect on the results of this experiment. The sex ratio of piglets we selected is 1:1, which more realistically reflects the changes in piglets before and after weaning in actual production.

 (2) Provided the experimental design, the appropriate analysis would actually entail a two-way ANOVA (simultaneous comparison among respective breeds & weaned vs. unweaned). Also, immunohistochemistry (and western blot?) data from all study groups were normalized to the unweaned Taoyuan Black group. Whether in the primary manuscript or supplement, the authors should also illustrate data as each breed of piglet normalized to itself (i.e., Taoyuan Black weaned vs. unweaned & Duroc weaned vs. unweaned) to best illustrate within-group comparisons among weaned and unweaned piglets among respective breeds.

Answer:

The previous answer shows the reason that a two-way ANOVA was not chosen. Based on your opinion, we have selected an independent T-test to count the data in the following studies and analyzed related data as each breed of piglet normalized to itself (Taoyuan Black weaned vs. unweaned & Duroc weaned vs. unweaned).

(3) Figures 2C, 2E, 3B, 4B, 4H, 4I, 4K, 5H, and S2B indicate "arbitrary units" but the data appear to be fold change units as normalized to a "control" group as 1. Please clarify and label accordingly.

Answer:

We revised fold change units as normalized to a CON group as 1.

(4) Figure panel labeling is not consistent, whereby by one letter will indicate several images/graphs within one Figure (e.g., Figures 3A & 3B) and not others (e.g., Figures 1A & 1B). Where reasonable, a recommendation is to just have each image/graph with its own letter label throughout Figures for consistency.

Answer:

We modified the letter label in each image.

Reviewer 2 Report

The study titled “Early weaning inhibits intestinal stem cell expansion to disrupt 2 the intestinal integrity of Durocpiglets via regulating the 3 Keap1/Nrf2 signalin” addresses a novel and important aspect of piglet health by exploring the impact of early weaning on intestinal stem cells (ISCs) and intestinal integrity, particularly comparing commercial Durocpiglets with local Taoyuan Black piglets. 

Strengths 

1. This comparative approach enhances the understanding of breed-specific differences in stress responses, which is valuable for optimizing piglet management strategies in the livestock industry. 2. The research question is well-defined, focusing on the ISC-driven epithelial renewal and the role of the Keap1/Nrf2 signaling pathway in weaning stress.  3. The study provides comprehensive data on multiple intestinal parameters, including villus height, tight junction-related proteins, and TEER values, which are crucial for understanding changes in gut integrity.  4. The focus on oxidative stress and the Keap1/Nrf2 pathway provides valuable mechanistic insights into how early weaning disrupts ISC function in Duroc piglets. 5- The manuscript presents data clearly, with appropriate figures and statistical analyses that support the study’s conclusions. The results are logically organized, making it easy for the reader to follow the experimental findings and their implications.

Minor comments 

1. While the study highlights differences between Duroc and Taoyuan Black piglets, the discussion lacks depth regarding the genetic or physiological reasons behind these variations. Including a more detailed comparison of the inherent stress resilience and genetic background of the two breeds would provide a better context for the observed differences. 2. The manuscript primarily focuses on oxidative stress and ISC markers but does not thoroughly explore inflammatory responses, which are often closely linked with oxidative stress and gut integrity. Including inflammatory cytokine analysis would provide a more comprehensive understanding of the intestinal response to early weaning 3. More detailed mechanistic studies, such as pathway inhibitors or activators, could strengthen the claims regarding the role of Keap1/Nrf2 signaling. 4. Minor grammatical errors and awkward sentence structures are present in the abstract and main text, which slightly affect the readability.  5. A thorough language review would improve the overall presentation of the manuscript.

Overall, this study provides significant insights into the breed-specific response of ISCs to weaning stress, emphasizing the role of the Keap1/Nrf2 pathway in intestinal integrity. Addressing the highlighted limitations would enhance the impact and clarity of the findings.

See above 

Author Response

List of reviewer comments and revisions of the manuscript

Dear Reviewers,

First of all, many thanks for your invaluable comments on our manuscript entitled “Early weaning inhibits intestinal stem cell expansion to disrupt the intestinal integrity of Duroc piglets via regulating the Keap1/Nrf2 signaling”. We have revised the paper according to your comments, as described below.

Thank you very much for your excellent comments on our manuscript. We were thrilled to revise our paper based on your comments. The changes are highlighted in RED in the revised manuscript.

The study titled “Early weaning inhibits intestinal stem cell expansion to disrupt 2 the intestinal integrity of Durocpiglets via regulating the 3 Keap1/Nrf2 signalin” addresses a novel and important aspect of piglet health by exploring the impact of early weaning on intestinal stem cells (ISCs) and intestinal integrity, particularly comparing commercial Durocpiglets with local Taoyuan Black piglets.

Strengths

  1. This comparative approach enhances the understanding of breed-specific differences in stress responses, which is valuable for optimizing piglet management strategies in the livestock industry.
  2. The research question is well-defined, focusing on the ISC-driven epithelial renewal and the role of the Keap1/Nrf2 signaling pathway in weaning stress.
  3. The study provides comprehensive data on multiple intestinal parameters, including villus height, tight junction-related proteins, and TEER values, which are crucial for understanding changes in gut integrity.
  4. The focus on oxidative stress and the Keap1/Nrf2 pathway provides valuable mechanistic insights into how early weaning disrupts ISC function in Duroc piglets.
  5. The manuscript presents data clearly, with appropriate figures and statistical analyses that support the study’s conclusions.

The results are logically organized, making it easy for the reader to follow the experimental findings and their implications.

Minor comments

1. While the study highlights differences between Duroc and Taoyuan Black piglets, the discussion lacks depth regarding the genetic or physiological reasons behind these variations. Including a more detailed comparison of the inherent stress resilience and genetic background of the two breeds would provide a better context for the observed differences.

Answer:

In the first paragraph of the discussion section, we described the characteristics (such as inherent stress resilience) of Taoyuan black pigs as native pigs. According to its stress resistance, we speculate that Taoyuan black pig can resist weaning stress. There is still a lack of research comparing the genetic background of Taoyuan black pigs with Duroc pig breeds. Thus, it makes sense to do comparative mining of gene information.

2. The manuscript primarily focuses on oxidative stress and ISC markers but does not thoroughly explore inflammatory responses, which are often closely linked with oxidative stress and gut integrity. Including inflammatory cytokine analysis would provide a more comprehensive understanding of the intestinal response to early weaning

Answer:

In this study, we did not detect the changes in inflammatory factors to explore the role of inflammatory factors in weaning stress-induced intestinal injury. Thus, we speculate that the different tolerances of pigs to weaning are possibly due to the differences in inflammatory factor levels or sensitivity to inflammatory factors before and after weaning based on the available literature. Related descriptions are presented in the discussion of the manuscript and marked in red.

3. More detailed mechanistic studies, such as pathway inhibitors or activators, could strengthen the claims regarding the role of Keap1/Nrf2 signaling.

Answer:

We performed early weaning of 16-day-old male mice and isolated the jejunal crypts in each group of mice to construct a model of early weaning injury in stem cells 3 days after weaning. Ex vitro treatment of intestinal organoids in the early weaning group with Nrf2 activator to alleviate weaning stress damage in intestinal stem cells. We describe methods for statistical indicators related to organoid amplification in detail in Materials and Methods 2.7. By processing 10 μmol/L Sulforaphane (Nrf2 activator), we found that up-regulation of Nrf2 signaling activity alleviated the inhibitory effect of early weaning on the expansion of mouse jejunal stem cells. We showed the results in the supplementary material in Figure S4. Thus, Nrf2 mediates early weaning-induced intestinal stem cell damage.

4. Minor grammatical errors and awkward sentence structures are present in the abstract and main text, which slightly affect the readability.

Answer:

We revised the grammatical errors and awkward sentence structure marked in red font.

5. A thorough language review would improve the overall presentation of the manuscript.

Answer:

The language of the manuscript has been revised and marked in red font.

Overall, this study provides significant insights into the breed-specific response of ISCs to weaning stress, emphasizing the role of the Keap1/Nrf2 pathway in intestinal integrity. Addressing the highlighted limitations would enhance the impact and clarity of the findings.

Round 2

Reviewer 1 Report

The authors have adequately addressed my original comments with the application of independent t-tests. However, instead of y-axis labeling as CON group 1, the authors should state Fold change of un-weaned group (or similar) while explaining in the legend that each respective un-weaned group was set at 1 for comparison relative to weaned.

See my comment above.

Author Response

Dear Reviewers,

First of all, many thanks for your invaluable comments on our manuscript, “Early weaning inhibits intestinal stem cell expansion to disrupt the intestinal integrity of Duroc piglets via regulating the Keap1/Nrf2 signaling.” We have revised the paper according to your comments, as described below.

We were thrilled to revise our paper based on your comments. The changes are highlighted in RED in the revised manuscript.

Major comments:

The authors have adequately addressed my original comments with the application of independent t-tests. However, instead of y-axis labeling as CON group 1, the authors should state Fold change of un-weaned group (or similar) while explaining in the legend that each respective un-weaned group was set at 1 for comparison relative to weaned.

Answer:

We state y-axis labeling as Fold change of Un-weaned group and describe in the legend and materials and methods marked in RED.